# Salicylic Acid in Plant Symbioses: Beyond Plant Pathogen Interactions

**DOI:** 10.3390/biology11060861

**Published:** 2022-06-03

**Authors:** Goodluck Benjamin, Gaurav Pandharikar, Pierre Frendo

**Affiliations:** 1Université Côte d’Azur, INRAE, CNRS, ISA, 06000 Nice, France; goodluck.benjamin@etu.univ-cotedazur.fr; 2Université de Lorraine, INRAE, UMR IAM, 54280 Champenoux, France; gaurav.pandharikar1@gmail.com

**Keywords:** salicylic acid, endophytes, nitrogen-fixing symbiosis, mycorrhizae, stress, microbes, symbiosis

## Abstract

**Simple Summary:**

Plants form beneficial symbioses with endophytes, arbuscular mycorrhizal fungi, and nitrogen-fixing rhizobia. In addition to their role in plant growth and development, these microorganisms enhance host plant tolerance to a wide range of environmental stress. Salicylic acid (SA) is widely known to play essential roles in plant defense against pathogens. In addition, SA has been shown to be involved in plant–microbe symbiotic interactions. In this review, we summarize the impact of SA on symbiotic interactions and on defense priming by beneficial microbes.

**Abstract:**

Plants form beneficial symbioses with a wide variety of microorganisms. Among these, endophytes, arbuscular mycorrhizal fungi (AMF), and nitrogen-fixing rhizobia are some of the most studied and well understood symbiotic interactions. These symbiotic microorganisms promote plant nutrition and growth. In exchange, they receive the carbon and metabolites necessary for their development and multiplication. In addition to their role in plant growth and development, these microorganisms enhance host plant tolerance to a wide range of environmental stress. Multiple studies have shown that these microorganisms modulate the phytohormone metabolism in the host plant. Among the phytohormones involved in the plant defense response against biotic environment, salicylic acid (SA) plays an important role in activating plant defense. However, in addition to being a major actor in plant defense signaling against pathogens, SA has also been shown to be involved in plant–microbe symbiotic interactions. In this review, we summarize the impact of SA on the symbiotic interactions. In addition, we give an overview of the impact of the endophytes, AMF, and rhizobacteria on SA-mediated defense response against pathogens.

## 1. Introduction

Plants have an innate immune system that detects and limits pathogen attacks. Pattern recognition receptors (PRRs) on the plant cell surface detect molecules containing characteristic patterns of microbes. Detection of these pathogen-/microbe-associated molecular patterns (PAMPs/MAMPs) leads to activation of pattern-triggered immunity (PTI). In many cases, PTI prevents further pathogen establishment. However, some pathogens have developed effector proteins that suppress PTI and therefore maintain pathogenicity. To resist pathogen-associated effector proteins, plants encode resistance proteins (R) to provide effector-triggered immunity (ETI). These R proteins, which are generally located within the plant cell, may directly or indirectly recognize their related pathogen- encoded effectors activating ETI [1,2]. Both PTI and ETI are associated with the activation of defenses in the infected tissue, including the generation of reactive oxygen species (ROS) [3], increase in intracellular Ca^2+^ concentrations, and activation of mitogen-activated protein kinases (MAPKs), to finally activate the expression of various defense-associated genes, synthesis of antimicrobial compounds, and accumulation of SA [4,5,6].

Usually, ETI induces plant defenses more rapidly and strongly than PTI. ETI is generally associated with the formation of necrotic lesion, which may help to restrict pathogen movement from the infection site. Subsequent to these local events, ETI and PTI can induce immune responses in the uninfected parts of the plant exposed to pathogen attack (Figure 1). This long-distance-induced broad-spectrum resistance is called systemic acquired resistance (SAR). SAR is primarily controlled by endogenous accumulation of salicylic acid (SA) and characterized by the activation of Pathogenesis-Related (PR) genes and proteins with antimicrobial activity [7,8,9]. Salicylic acid (SA; 2-hydroxybenzoic acid) is a critical hormone that plays direct or indirect roles in regulating many aspects of plant growth and development as well as thermogenesis, drought resistance, and disease resistance [10].

SA is linked to several key components of plant defense through intricate networks, generally activated following infection by biotrophic pathogens, which require living host tissue. SA acts through the non-expressor of PR1 protein (NPR1), a pivotal component in plant defense signaling. The detailed mechanism of SA-mediated regulation of defense through NPR1-mediated signaling and its regulation in the cytoplasm and nucleus has been well documented [11,12,13]. In Arabidopsis, proteins NPR1 to NPR6 constitute a multigenic family [14]. Among these, NPR3 and NPR4 have been shown to interact with SA [15,16], functioning as adaptors of the Cullin-3 ubiquitin E3 ligase to mediate NPR1 degradation in an SA-regulated manner [17]. In addition, NPR3 and NPR4 have also been shown to bind SA directly to modulate their interaction with NPR1 [17]. NPR1 is constitutively expressed in most cell types, and it remains mostly inactive in the cytosol in an oligomeric form until the host is infected with a pathogen. After infection, the host plant produces more SA, and the higher SA content is associated with an alteration in the cellular redox potential. As a result of this, the NPR1 oligomer is reduced to biologically active monomers. The monomeric NPR1 then moves to the nucleus where it interacts with TGA proteins. This interaction results in the expression of SA-dependent pathogenesis-related (PR) genes. Since PR are conserved proteins in the plant kingdom, the expression of their genes is frequently used as a robust marker to determine and characterize the SA-response during the acquired SAR [18,19].

In addition to SA, jasmonic acid (JA)- and ethylene-signaling pathways are usually required for the activation of plant defense against necrotrophic pathogens and herbivores [20]. JA is also commonly associated with induced systemic resistance (ISR) (Figure 1). ISR is an improved resistance of the host plant triggered by beneficial microbes [21]. Plant defense response is energy-consuming, and extensive crosstalk between JA- and SA-mediated defense signaling pathways occurs to efficiently allocate energy and provide robustness to the plant immune system [22].

In addition to being a key player in plant disease reaction, SA has also been reported to be involved in interactions between plants and beneficial microbes. Interactions between plants and beneficial microbes include multiple symbiotic relationships. Plant endophytes are microbial organisms (bacteria or fungi), which live within the plant without causing apparent disease (Figure 2). Endophytes assist their host by transformation and solubilization of nutrients and other micro minerals. In addition, they induce the tolerance response of the host against stress caused by abiotic factors such as osmotic stress and exposure to heavy metals and xenobiotics. Apart from abiotic factors, endophytes also assist their host plant in the suppression of harmful microorganisms and act in the biological control against plant pathogens [23]. Plants may also interact with mycorrhizal fungi, resulting in an improved plant nutrition and an induced resistance to a number of pathogens or abiotic stresses [24,25]. Finally, another symbiotic interaction of interest which allows for better plant nutrition in a nitrogen-deficient environment is the nitrogen-fixing symbiosis (NFS). In NFS, the beneficial relationship established between plants and soil bacteria results in root nodule formation and biological atmospheric nitrogen (N_2_) fixation. Both partners benefit from this interaction, as the host plant is provided with a ready source of fixed nitrogen, and in exchange bacteria receive carbon source for development and protection from environmental conditions [26].

All these relationships involve an intimate crosstalk between the plant and the symbiotic microorganisms. The establishment and the maintenance of symbiosis should be tightly controlled by the plant to balance energy cost with benefits. In this context, the autoregulation of both mycorrhizae (AOM) and nodulation (AON) are involved in the regulation of the interaction to avoid an over-infection of the plant [27]. Moreover, the intracellular invasion of the nodule cells by the rhizobia also involves specific control of the plant immunity in root nodules to allow intracellular colonization of the plant cell [28,29]. This review focuses on the role of SA in plant–microbe symbiotic interactions, including endophytes, mycorrhiza, and nitrogen-fixing bacteria. We also discuss how SA might be involved in the priming of the defense of symbiotic plants against pathogens.

## 2. SA in Plant–Endophyte Interactions

Endophytes are microorganisms that can live in the internal parts of plants. Endophytes colonize their host plants without causing any symptoms [30,31,32,33]. The intimate association of endophytic microbes with the plant tissues improves plant survival and productivity through nutrient acquisition [34]. Moreover, it increases the plant’s ability to tolerate abiotic stress and decrease biotic stress by enhancing plant resistance to bio-aggressors such as pathogens, insects, and herbivores [35,36,37].

Endophytes colonize the extracellular space of living plant tissue. Thus, they can potentially interact with the defensive signaling pathways of their host. Navarro-Melendez and Heil investigated whether the interaction of Lima bean (*Phaseolus lunatus*) with *Bartalinia pondoensis* (C015), *Fusarium* sp. (U090) and *Cochliobolus lunatus* (U065) alters the endogenous levels of SA and JA and the expression of the JA-dependent indirect defense traits (extrafloral nectar secretion and volatile organic compounds emission) [38]. The authors found that plant SA level was significantly decreased in plants by all of the strains [38]. Moreover, their results suggest that the endophytes interact in complex and strain-specific ways with the endogenous levels of SA and JA and with the defense traits that are controlled by these hormones. These findings are in line with the hypothesis that the endophytes decrease endogenous levels of SA, likely due to SA–JA trade-offs [39].

The mechanism behind the SA down accumulation has not been fully investigated. However, bacteria contain salicylate hydroxylase, nahG, which are able to metabolize SA in an inactive molecule [40,41]. In studying SA degradation in endophytic fungi, Graminha and colleagues reported that *Aspergillus nidulans* contained a salicylate hydroxylase-like gene which was able to catabolize the naphthalene portion of terbinafine in a similar manner as do bacteria [42]. This ability to degrade SA seems to be a common phenomenon among fungi [43]. However, Ambrose and colleagues characterized the endophytic fungi *Epichloë festucae* salicylate hydroxylase in red fescue [44]. They found that *in planta* expression of the *E. festucae* salicylate hydroxylase did not significantly modify the SA level in endophyte-infected plants compared to endophyte-free plants. This suggests that the expression of salicylate hydroxylase by the endophyte was not the main factor in the lack of a host defense response during endophyte colonization [44]. This further strengthens the hypothesis that SA- and JA-mediated defense response pathways are mutually antagonistic in plant–endophyte interactions [45].

The study of Schmid and colleagues analyzed the transcriptomes of *Epichloë festucae* and its host *Lolium perenne* in host tissues of different function and developmental stages [46]. They found 289 genes that are induced more than two times in endophyte-inoculated plants compared to uninoculated plants. Among these, 85 are associated with hormone synthesis, transport, and metabolism showing that the symbiosis strongly affects the plant hormonal homeostasis. The alteration in gene expression associated to hormones is linked to SA, JA, gibberellin, ethylene, abscisic acid, cytokinin, and auxin. However, no specific link was observed with SA and no *PR* genes were found in the defense genes induced more than twice in inoculated plants, suggesting that the modification of the plant defense metabolism is not significantly associated with SA in this symbiotic interaction [46].

To understand the impact of SA on endophyte colonization of plants and its further consequence on the response of plants to herbivores, the effect of the exogenous application of SA was studied [47]. It was shown that plants in symbiosis with *Epichloë* fungal endophytes had lower concentrations of SA than did endophyte-free plants. Upon the exogenous application of SA, physiological concentration of the SA was highly increased and a significant reduction in endophyte-produced alkaloid (loline) was observed in the leaves. This reduction was correlated to an increased susceptibility to aphids [47]. These results support the hypothesis that high SA content suppresses some fungal endophytes [48].

Endophytes also protect plants through the priming of plant defense mechanisms. In this context, *Paenibacillus alvei* K165 was able to protect *Arabidopsis thaliana* against the fungi *Verticillium dahlia* [49]. The priming reaction was associated with the induced expression of *PR1*, *PR2,* and *PR5* genes. Moreover, the defense reaction further required intact SA biosynthesis and signaling pathways [49]. Likewise, root-colonizing *Pseudomonas fluorescens* strain SS101 increased resistance in *A. thaliana* against *Pseudomonas syringae* pv tomato (Pst) and the insect pest *Spodoptera exigua*. Transcriptomic analysis and bioassays with specific *Arabidopsis* mutants showed that the induced resistance response to Pst was dependent on SA and not on JA and ethylene [50]. Furthermore, the involvement of SA in the priming of plant defense by *Trichoderma* strains against *Botrytis cinerea* was described in *Arabidopsis* and *Solanum lycopersicum* [51,52,53]. Finally, Kou and colleagues showed that *Epichloë* endophytes are capable of inducing SA-dependent defense responses in the host plants to provide significant resistance against the pathogen *Blumeria graminis* [54].

Taken together, these studies indicate that plant–endophyte interaction could be modulated by SA level (Table 1). Moreover, plant–endophyte interaction could also modulate the SA metabolism among other hormones and prime plant defense reaction against bio-aggressors. However, this is not a general conclusion, as JA signaling was also shown to be associated with increased plant defense metabolism depending on the symbiotic partners.

## 3. SA in the Plant-Mycorrhiza Interactions

Mycorrhiza is a mutualistic symbiotic association between plants and root-colonizing fungi. During this association, the fungi provide nutrients to the plant in exchange for photosynthetic products [55]. In addition to their involvement in the nutrient supply, mycorrhizal fungi increase plant ability to tolerate abiotic and biotic stress [56].

Numerous studies indicate that in the plant host/Arbuscular mycorrhiza fungi (AMF) interaction, plant defense responses are induced during early root colonization and are repressed subsequently [57,58]. A transient accumulation of SA during the first stage of AM root colonization, associated with the early induction of plant defense responses, has been reported [59,60]. The role of SA in the regulation of root colonization has been suggested during the establishment of the AM symbiosis. Indeed, the inability of Myc^-^ Pisum sativum mutants to form the AM association was linked to enhanced SA level during the early steps of the interaction [61].

**Table 1 biology-11-00861-t001:** Involvement of salicylic acid in plant–microorganism symbiotic interactions and defense priming.

Interaction	Activity	Effect	Microbe	Host Plant	Ref.
Plant-Endophyte	Establishment of symbiosis	Downregulation of SA accumulation	*Bartalinia pondoensis, Fusarium* sp., *Cochliobolus lunatus*	*Phaseolus lunatus*	[38]
			*Epichloë* spp.	*Lolium multiflorum*	[47]
	Defense	Antiherbivory	*Epichloë spp*.	*Lolium multiflorum*	[47]
		Enhanced VOC emission	*Bartalinia pondoensis, Fusarium* sp., *Cochliobolus lunatus*	*Phaseolus lunatus*	[38]
		Induction of *PR* genes.	*Paenibacillus alvei*	*Arabidopsis thaliana*	[49]
		Induction of defense related genes	*Pseudomonas fluorescens*	*Arabidopsis thaliana*	[50]
			*Trichoderma* spp.	*Arabidopsis thaliana*	[51,52]
				*Solanum lycopersicum*	[53]
			*Epichloë gansuensis*	*Achnatherum inebrians*	[54]
Plant-Mycorrhiza	Establishment of symbiosis	Regulation of root colonization	*Glomus mosseae*	*Pisum sativum*	[61]
			*Glomus intraradices, Glomus mosseae*	*Nicotiana tabacum*	[62]
			*Funneliformis mosseae, Rhizophagus irregularis*	*Solanum lycopersicum, Glycine max, Zea mays*	[63]
			*Glomus* sp.	*Solanum tuberosum, Medicago truncatula*	[64]
	Defense	Defense Priming	*Glomus mosseae*	*Trifolium repens*	[65]
			*Rhizofagus irregularis*	*Vitis vinifera*	[66]
		Induction of defense related genes	*Claroideoglomus etunicatum, Claroideoglomus claroideum, Rhizophagus irregularis, Funneliformis geosporus, Funneliformis mosseae*	*Pisum sativum*	[67]
			*Glomus* sp.	*Solanum tuberosum, Medicago truncatula*	[68]
			*Funneliformis mosseae*	*Triticum aestivum*	[69]
			*Glomus intraradices*	*Oryza sativa*	[70]
			*Glomus mosseae*	*Oryza sativa*	[59]
Plant-Rhizobia	Establishment of symbiosis	Regulation of root colonization and nodule formation	*Mesorhizobium loti*	*Lotus japonicus, Medicago truncatula*	[71]
			*Sinorhizobium meliloti*	*Medicago sativa*	[72]
			*Sinorhizobium meliloti*	*Medicago sativa*	[73]
		Decreased innate immunity within nodules	*Sinorhizobium* spp.	*Medicago truncatula*	[74]
			*Sinorhizobium* spp	*Medicago truncatula*	[75]
	Defense	Induction of defense related genes	*Sinorhizobium* spp.	*Medicago truncatula*	[75]
			*Rhizobium leguminosarum, Sinorhizobium meliloti*	*Medicago truncatula, Pisum sativum*	[76]
			*Rhizobium leguminosarum*	*Pisum sativum*	[77,78]
			*Sinorhizobium meliloti*	*Medicago truncatula*	[79]

Moreover, SA exogenously applied to roots in AMF–rice interaction reduced root colonization at the early stages of the interaction [59]. However, no effect of SA on AMF appressoria formation was observed [59]. Application of SA to leaves of cucumber plants showed no effect on the interaction with AMF excluding its systemic effect on the symbiotic interaction [80]. The effect of SA level on plant–mycorrhizae interactions was confirmed by Herrera-Medina and colleagues [62]. They showed that NahG tobacco plants with lower SA levels showed higher levels of root colonization, more infection units, and more arbuscules [62]. In contrast, constitutive SA biosynthesis (CSA) tobacco plants inoculated with Glomus intraradices or G. mosseae showed lower root colonization. The result of the SA level on mycorrhization was similar for the infection with two AMF strains showing that the effect was not specific to a fungal strain [62].

To better understand the defense reaction occurring in the plant–AMF interaction, an integrative analysis of the response of phylogenetically diverse plants—tomato, soybean, and maize—to two mycorrhizal fungi—Funneliformis mosseae and Rhizophagus irregularis—was carried out [63]. The two AMF had different influence on the levels of SA, and the associated transcriptional response was dependent on the plant and mycorrhizal fungi species. F. mosseae induced the SA-related pathway in the tomato, while R. irregularis had no effect. In contrast, SA accumulation was induced by R. irregularis root colonization in maize. Although SA level increases have been shown to be related to the initial stages of the AM interaction [81], Fernández and colleagues demonstrated that higher SA levels are associated with later stages of AM symbiosis in tomato and maize [63]. The variation in SA levels depends on the partners’ genotypes, and more specifically on the colonizing fungus [63]. Similarly, elevated levels of SA have been also reported in G. mosseae colonized clover [65] and barley [82]. In grapevine, SA level was significantly higher in leaves of plants colonized with Rhizofagus irregularis at 2 months after mycorrhization. In contrast, mycorrhizal grapevine roots contain significantly lower levels of SA [66]. It has been proposed that SA signaling has a biphasic induction during AM symbiosis, with a first increase in pre-symbiotic stages that level off as the colonization initiates, and a second induction at later stages of root colonization likely to control colonization extension [64,83].

The mycorrhiza symbiosis has been reported to have a priming effect against some pathogens [84,85,86]. The protective effect of mycorrhiza on below-ground pathogens was correlated with a decrease in pathogen content within the plant tissues, which may be linked to the competition for space between the pathogen and the mycorrhizal fungi. However, the modification of root metabolism during the interaction with AMF can also increase the plant protection through the increase in defense compounds [60,65,87,88]. In clover inoculated with G. mosseae, the increase in both free and cell wall–bound phenolics was associated with the activation of phenylalanine ammonia-lyase (PAL) activity and the accumulation of nitric oxide (NO), hydrogen peroxide (H_2_O_2_), and SA [65]. This highlights that SA level may be critical in the phenolic synthesis. The effect of mycorrhiza on the induction of plant defense against leaf pathogens was also observed multiple times [68,69,70,87]. Using metabolomics and proteomics approaches, Sistani and colleagues studied the impact of AMF colonization on pathogen resistance in pea (Pisum sativum) against Didymella pinodes [67]. They demonstrate that AMF improves seed yield and protects two pea cultivars, Protecta and Messire, upon pathogen attack. However, the plant protection was different depending on the plant cultivar, with a more effective response to pathogen attack in the Protecta cultivar, showing genotype-specific defense strategies. Moreover, the defense priming effect seems to be associated with JA accumulation and defense induction as proteins involved in JA synthesis, N-jasmonoyl isoleucine and JA-responsive proteins, are significantly accumulated in mycorrhized peas compared to non-mycorrhized plants [67]. Besides the involvement of JA, the hormones SA, and abscisic acid, ethylene was also potentially associated with resistance response [89]. The crosstalk between the different hormonal pathways including SA remains to be analyzed in the different multitrophic interactions to understand the mechanisms involved in the long-distance regulation of plant defense against bio-aggressors.

In contrast to the induced defense by mycorrhiza, multiple reports showed that mycorrhization may have a negative effect on plant defense against some pathogens [90]. These studies show that mycorrhiza-mediated beneficial or negative effects are dependent on the species of plant, symbiotic fungi, and pathogen.

## 4. SA in Plant–Rhizobia Symbiosis

Nitrogen is an essential macronutrient for plants which determines growth, development, and yield of plants [91]. In nitrogen-deficient soils, legume and actinorhizal plants have the ability to establish symbiosis with soil bacteria and form nodules where the bacteria obtain the appropriate conditions to reduce atmospheric nitrogen into ammonia [92]. Both bacterial–plant partners benefit from this symbiotic relationship. Plants receive a ready source of reduced nitrogen from bacteria and in exchange bacteria receive protected environment and usable carbon sources from their host plant [93].

In legume–rhizobia symbiosis, the infection of the plant by compatible rhizobia is usually associated with a transient activation of plant defence followed by its reduction which allows an efficient invasion of the plant cells by thousands of bacteroids [74]. The defence response could be induced during some rhizobia–plant interactions and has been suggested to play an important role in determining host range or in controlling the presence of the bacteria in the plant [94,95]. Impaired nodulation of alfalfa with a nodC mutant of *Sinorhizobium meliloti* unable to synthesize the lipochitin Nod signal required for infection was associated with an increased accumulation of H_2_O_2_ and SA in roots, suggesting a defense reaction in the inefficient interaction [72,96]. Similarly, an induced SA-dependent defense reaction was associated with an impaired nodulation elicited by glutathione deficiency [97]. Thus, SA is linked to defense induction in impaired legume–rhizobium interactions.

It is widely accepted that leguminous plants have the ability to regulate nodulation, and autoregulation occurs at different developmental stages [98,99]. This autoregulation may be associated with a localized defense response. These responses could potentially be involved in limiting nodulation during the infection process, such as infection thread and nodule formation. Vasse and colleagues reported that a plant defense response was associated with localized root cell necrosis and accumulation of phenolic compounds [73]. This defense reaction could be involved in the abortion of the infection threads during infection of alfalfa by *Sinorhizobium meliloti* and reduce the successful infection [73]. Exogenous SA treatment on the root resulted in both reduced and delayed nodule formation on alfalfa or vetch roots, reinforcing the idea that SA regulates the early steps of the root infection by the rhizobia [72,100]. To analyse the importance of endogenous SA, Stacey and colleagues constructed transgenic *L. japonicus* plants expressing salicylate hydroxylase, encoded by the bacterial *NahG* gene and *NahG*-expressing *Medicago truncatula* roots [71]. These *NahG* transgenic *L. japonicus* plants presented significantly lower accumulation of endogenous SA associated with higher nodulation level. The nodulation was also increased for transgenic *M. truncatula* roots, but it was not possible to detect a significant reduction in the SA content [71]. Taken together these reports indicate that SA-mediated plant defense pathways are involved in modulating legume infection both during indeterminate and determinate nodulation.

A recent study suggests that innate immunity is reduced or suppressed within nodules [75]. This reduction likely enables colonization and nodulation process and maintains viable rhizobia populations inside the plant. Benezech and colleagues evaluated the potential consequences and risks associated with an altered immunity in the symbiotic organ [75]. They used a tripartite system with the model legume *Medicago truncatula*, its nodulating symbiont of the genus Sinorhizobium, and the pathogenic soil-borne bacterium *Ralstonia solanacearum*. They also observed various colonization patterns for nodules, suggesting that the pathogen can enter the nodules through multiple ways and that nodule innate immunity was altered in the symbiotic organs allowing an easier nodule infection by the pathogens. However, defense gene expression was activated in nodules and roots in response to *R. solanacearum* infection. *R. solanacearum*-induced genes showed the same expression kinetics in roots and nodules, notably genes associated with the SA defense pathway, indicating that nodules can activate defense reactions upon infection with *R. solanacearum*. Interestingly, 130 defense-related genes were found specifically induced in *R. solanacearum*-infected nodules and not in roots, indicating that nodules develop defense reactions distinct from roots as a whole [75].

In parallel with nodule local defense, SA has also been associated with defense priming by nitrogen-fixing symbiosis. Rhizobium inoculation decreases disease severity levels against *Didymella pinodes* by lower pathogen infection, significant reduction in seed infection level, and higher accumulation of the phytoalexin pisatin [77,78]. To investigate whether symbiotic rhizobia are able to modulate resistance against biotrophic pathogens, Smigliesski and colleagues analysed the impact of pre-established nodulation of *M. truncatula* and *P. sativum* against the powdery mildew fungus *Erysiphe pisi* [76]. In *M. truncatula*, nodulation resulted in a reduced penetration by *Erysiphe pisi* with a similar degree of sporulation in nodulated and non-nodulated plants. In contrast, nodulation of the pea did not affect fungal penetration but resulted in a significant reduction in sporulation. This result indicates that nodulation of *M. truncatula* and *P. sativum* improves early pre-penetration (*M. truncatula*) and late post penetration (pea) resistance against *E. pisi* [76]. Moreover, *E. pisi* inoculation of *M. truncatula* significantly increased the concentration of free SA in leaves of nodulated plants at both 1 and 7 dpi. By contrast, significant increase was observed only at the later time in non-nodulated plants [76]. Similarly, the concentration of free SA was significantly increased in nodulated peas at 7 dpi. These findings show that nodulation of pea and *M. truncatula* primed the plants for *E. pisi*-triggered accumulation of free SA that correlated with increased resistance against the powdery mildew fungus. It should be noted that in the same context, nodulation did not protect *M. truncatula* from *Xanthomonas campestris pv alfalfa* [76].

Pandharikar and colleagues demonstrated that nitrogen-fixing symbiosis influences plant–aphid interactions and the plant’s defense response [79]. They observed a detrimental effect of rhizobia-inoculated plants on the aphid development as they had lower weight compared to aphids that fed on nitrate-supplemented plants. In addition, *PR1* expression was strongly upregulated in infested plants, confirming the activation of SA-dependent defense; but a higher expression of *Proteinase Inhibitor* gene, a gene marker for JA transduction pathway, was observed in the leaves of nodulated plants [79].

All together, the results point to the regulatory role of the SA-mediated pathway in nitrogen-fixing symbiosis and to the defence-priming role of NFS either through SA or JA regulation pathways in interactions with plant bio-aggressors.

## 5. The Future Challenges of Hormonal Regulation in Plant-Beneficial Microbe Interactions

The involvement of SA in the interaction between plants and beneficial microbes has been clearly demonstrated in recent years. However, the effects of the modulation of the plant SA metabolism by the beneficial microbes remain largely elusive. Multiple reports showed that the modification of SA metabolism was associated with increased plant resistance to biotic and abiotic stress. In this context, the molecular mechanisms underlying this improved adaptation of the plants to the environment remain open questions.

The heterogeneity of the resulting effect of the plant-beneficial microbes in term of plant response to the microbes is also an intriguing topic. This heterogeneity suggests that plant and microbial genetic variability play a significant role in the results of the interaction. In this context, the search for plant and microbe genes regulating the outcome of the interaction through screening of the genetic diversity associated with the success of the interactions seems an exciting line of research. This screening for adapted species which present better interactions is also attractive for increasing the use of microbes for bio-stimulation and biocontrol in agroecological practices. Indeed, the use of beneficial microbes will be helpful to improve plant growth in impaired plant growth conditions or a reduced use of fertilizers and pesticides.

## Figures and Tables

**Figure 1 biology-11-00861-f001:**
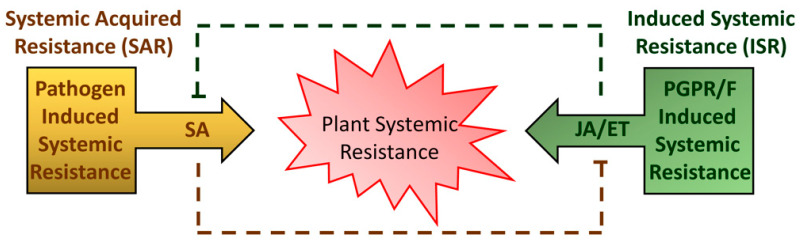
Scheme of different types of plant systemic resistance. The systemic acquired resistance (SAR) is triggered upon pathogen attack. Local defenses are followed by the production of mobile signals—mainly salicylic acid (SA)—that prime distal plant parts for defense compounds accumulation. Induced systemic resistance (ISR) can be triggered by colonization with plant-growth-promoting rhizobacteria or fungi (PGPR/F). ISR is regulated mainly by jasmonic acid (JA) and ethylene (ET).

**Figure 2 biology-11-00861-f002:**
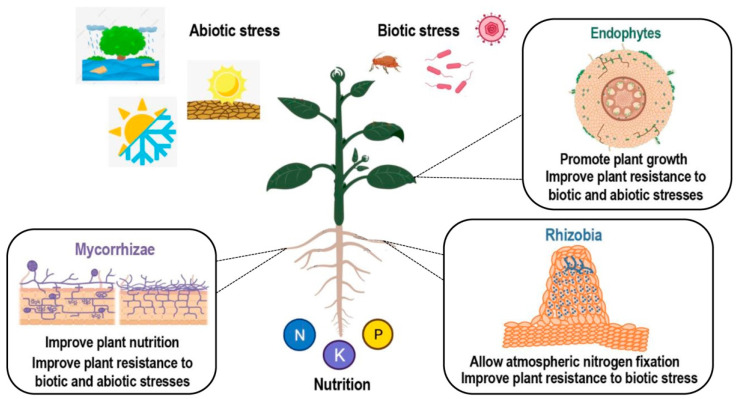
Scheme of different types of plant–microorganism symbiotic interactions. Positive effects of endophytic, arbuscular mycorrhizal (AM), and rhizobial colonization. Benefits from colonization include improved nutrition and tolerance to many abiotic and biotic stresses. Created with BioRender.com.

## Data Availability

Not applicable.

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
