# Peer review of "Salicylic Acid in Plant Symbioses: Beyond Plant Pathogen Interactions"

_biology, 2022, doi:10.3390/biology11060861_

Round 1

Reviewer 1 Report

MDPI, Biology. Review: Salicylic Acid in Plant Symbioses: Beyond Plant Pathogen Interactions. Benjamin et al. 

The review focuses on an interesting and timely topic and meets MDPI Biology standards. The review is nicely written, however I would love to see more details about salicylic acid interactions with specific genes in plants metabolic pathway. Also, interaction of SA with CO2 and ROS.

I would recommend to add following references:

- https://doi.org/10.1007/s42729-022-00884-y

- https://doi.org/10.1016/j.apsoil.2020.103710

- https://doi.org/10.1094/MPMI-05-22-0104-R

- https://doi.org/10.3390/ijms23105568

As it stands, the manuscript is ready to be published.

One specific recommendation:

- line 91: add 'in' to the 'to be involved the interactions'

Author Response

We thank the reviewer for the comments and the references. We added the references 

- https://doi.org/10.1007/s42729-022-00884-y (ref 34)

- https://doi.org/10.1016/j.apsoil.2020.103710 (ref 36)

  • https://doi.org/10.3390/ijms23105568 (ref 3)

We modify the line 91 (now 103)

Reviewer 2 Report

The manuscript “Salicylic Acid in Plant Symbioses: Beyond Plant Pathogen Interactions” by Benjamin et al. summarizes the role of SA on the symbiotic interactions, with special attention to its role mediating defense response against pathogens.

The article is well written and reads very well. Overall, the manuscript well summarizes current knowledge of the role of AS in symbiotic interactions and how it mediates the defense response against pathogens.

The topic is very interesting and of great significance to the scientific community working on the study area of plant protection and plant-microbe interactions.

I have no comments or suggestions that could improve this review. Furthermore, I believe that it is ready to be published in its present form.

Author Response

We thank the reviewer for the positive comments.

Reviewer 3 Report

The work of Benjamin and colleagues is a good review work about the role of salicylic acid, and other phytohormones, involved in plant-microbe interactions, particularly with microbial endophytes.

The manuscript (ms.) has a good level of English, though I am not a native English speaker; figures are also good and very explanatory.

My only suggestion is to include more recent works in their work, since many of them are very old, I mean, it is good to cite pioneering works, but the author should include novel works, from 2019 to today (some 10 to 15 works). There are plenty of them in the literature!

Author Response

We thank the reviewer for the positive comments. We added references corresponding to novel works in the manuscript (references in yellow).